# Unique Growth Pattern Presentation of a Papillary Renal Cell Carcinoma

**DOI:** 10.3390/diagnostics12081904

**Published:** 2022-08-06

**Authors:** Octavia Oana Harich, Gheorghe-Emilian Olteanu, Ioana Maria Mihai, Marius Benta, Gavriliuc Oana Isabella, Paunescu Virgil, Florina Maria Bojin

**Affiliations:** 1Department of Functional Sciences, “Victor Babes” University of Medicine and Pharmacy Timisoara, Eftimie Murgu Sq. No. 2, 300041 Timisoara, Romania; 2Department of Infectious Diseases, Discipline of Pulmonology, Center for Research and Innovation in Personalized Medicine of Respiratory Diseases, “Victor Babes” University of Medicine and Pharmacy, Timisoara Eftimie Murgu Sq. No. 2, 300041 Timisoara, Romania; 3Center of Expertise for Rare Lung Diseases, Clinical Hospital of Infectious Diseases and Pneumophthisiology “Dr. Victor Babes” Timisoara, Gh. Adam Street No. 13, 300310 Timisoara, Romania; 4Department of Microscopic Morphology, Discipline of Morphopathology, Anapatmol Research Center, “Victor Babes” University of Medicine and Pharmacy Timisoara, Eftimie Murgu Sq. No. 2, 300041 Timisoara, Romania; 5Timis County Emergency Clinical Hospital “Pius Brînzeu” Timisoara, No. 156 Liviu Rebreanu, 300723 Timisoara, Romania; 6XVision AI, Calea Torontalului No. 69, 021923 Timisoara, Romania; 7Department of Functional Sciences, Immuno-Physiology and Biotechnologies Center, “Victor Babes” University of Medicine and Pharmacy, Eftimie Murgu Sq. No. 2, 300041 Timisoara, Romania; 8Timis County Emergency Clinical Hospital “Pius Brînzeu” Timisoara, Center for Gene and Cellular Therapies in the Treatment of Cancer—OncoGen, No. 156 Liviu Rebreanu, 300723 Timisoara, Romania

**Keywords:** papillary renal cell carcinoma, growth pattern, IHC, NGS

## Abstract

Papillary renal cell carcinoma (PRCC) is defined by the WHO 2022 classification as a malignant tumor derived from the renal tubular epithelium. However, the WHO 2016 classification subdivided PRCC into two types, with type 1 PRCC showing papillae covered by a single layer of neoplastic cells, and type II PRCC, which can show multiple types of histologies and is more aggressive. The WHO 2022 classification eliminated the subcategorization of PRCC. Here, we present a histopathological case study with a 4-year follow-up diagnosed in 2018 as type I PRCC (WHO 2016) with intra-pyelocalyceal growth pattern in a 59-year-old male patient with a history of Type II diabetes mellitus, left-sided renal–ureteral lithiasis, and benign hypertrophy of the prostate. Microscopically the tumor was composed of small cuboidal cells with inconspicuous nucleoli, arranged on a single layer of tubulo-papillary cores, and scant, foamy macrophages. The tumor had a non-infiltrative, expansive pyelocalyceal growth pattern. Immunohistochemically (IHC), the tumor cells were CK7-intense and diffusely positive, and stained granular for AMACR. Next-generation sequencing (NGS) was performed for the tumor and the normal adjacent tissue for in-depth pathological characterization. To our knowledge, this is the first reported case where a PRCC displays this unique intra-pyelocalyceal growth pattern, mimicking a urothelial cell carcinoma of the renal pelvis system.

## 1. Introduction

Papillary renal cell carcinoma (PRCC) is mainly associated with end-stage renal disease (ESRD) with scarring or acquired cystic disease. Even if rare, some genetic syndromes are associated with PRCC, e.g., Birt–Hogg–Dubé syndrome. In addition, the hereditary PRCC syndrome presents with a high degree of penetrance in the families affected by it [1]. An in-depth molecular characterization analysis, published by the Cancer Genome Atlas Research Network (TCGA) [2], showed that type 1 and type 2 PRCC are not only histologically distinct but are also biologically and clinically divergent. Type 1 PRCC is associated with tyrosine-protein kinase (MET) pathway alterations, while type 2 PRCC is linked with activation of the nuclear factor erythroid 2-related factor 2/antioxidant responsive element (Nrf2-ARE) pathway. Furthermore, type 2 PRCC can be subdivided into at least three subtypes, based on molecular and distinctive phenotypic features [2].

PRCC remains the second most commonly encountered morphotype of renal cell carcinoma (RCC), with clear-cell renal-cell carcinoma (ccRCC) being the most common. Overall, when compared with clear-cell RCC, PRCC presents with a significantly higher rate of organ-confined tumor staging (pT1-2N0M0), and a higher five-year cancer-specific survival (CSS). In most of the cases, type 1 PRCC tumors show an exophytic spherical growth pattern, with pseudo-necrotic changes and/or the presence of a pseudo-capsule, all of which can be considered to be typical signs of a type 1 PRCC [3]. The histopathological review of PRCC presents a high degree of heterogeneity with regards to PRCC type 2, which can present with multiple types of histologies as opposed to PRCC type 1 which has a distinct specific histology and is generally diagnosed on hematoxylin and eosin (H&E) stains alone [2,3]. In 2022, in the fifth edition of the WHO Classification of Tumors of the Urinary and Male Genital Systems [4] the subcategorization of PRCC was eliminated and a single malignant entity was maintained, i.e., PRCC with no further divisions into type 1 or type 2. This elimination was implemented because of the frequent identification of tumors with mixed histologies (i.e., phenotypes), and the tumor entities that subdivided the type 2 PRCC category, because of this, from hereafter, the tumor in our case will be referred to as a PRCC.

Here, we present the detailed diagnosis and DNA-sequencing profile of a PRCC that displays a unique (imaging and gross) intra-pyelocalyceal growth pattern, mimicking a urothelial cell carcinoma of the renal pelvis.

## 2. Case Presentation

A 59-year-old male patient with a history of Type II diabetes mellitus, renal–ureteral lithiasis, and benign prostate hypertrophy was admitted to the emergency room with macroscopic hematuria. Following a urologic computed tomography (CT) scan, the patient was diagnosed with a right renal mass measuring 3.6 cm in diameter, with a possible diagnosis of urothelial carcinoma of the renal pelvis (Figure 1). Next, a flexible ureteroscopy was performed with visualization of the urinary bladder, the entire tract of the right ureter, which revealed normal-appearing mucosa, and the right renal pelvis where a reddish mass was identified; the mass was described as difficult to ascertain for positioning due to difficulties in acquisition and visualization.

A surgical intervention for a right total nephrectomy was chosen as the treatment of choice by the surgical department, with follow-up surgical reintervention if needed. The patient was assigned to the high-risk category of upper-urinary tract urothelial-cell carcinoma (UUTC), based on the 2018 European Association of Urology (EAU) guidelines [5], because of the tumor size of more than 2 cm and signs of hydronephrosis. No cytology or histology confirmation was performed before the surgical intervention.

The pathological grossing of the kidney revealed a small nodular lesion located medially, near the corticomedullary boundary, extending in a cordon-like manner from the minor calyces to the ureteropelvic junction and proximal ureter (Figure 1). The H&E stain showed microscopically that the tumor was composed of small cuboidal cells with inconspicuous nucleoli, arranged on a single layer of tubulo-papillary cores. The tumor stroma was noted for scant foamy macrophages (Figure 1). The tumor had a non-infiltrative, expansive pyelocalyceal growth. Immunohistochemically (IHC), the tumor cells were CK7-intense and diffusely positive, stained granular for alpha-methyl acyl-CoA racemase (AMACR), (Figure 1), and negative for Wilms tumor protein (WT1) (not shown).

After the pathology report was completed with a diagnosis of type I PRCC WHO ISUP (2016)/PRCC WHO ISUP (2022)—G1, pT1b, one representative slide was chosen, and two areas were marked; one tumor area and one histologically normal area. Next, the corresponding formalin-fixed paraffin-embedded (FFPE) tissue block was used to extract, with the use of a 0.1 mm punch biopsy instrument, the corresponding areas marked on the H&E slide, and the two extracted cores were subsequently processed for NGS. NGS was chosen to try to clarify the mutational background underpinning the unique growth pattern seen in the tumor.

Briefly, the following protocol was used for the NGS: (1) DNA purification—GeneJET Genomic DNA Purification Kit (ThermoFisher Scientific Inc., Waltham, MA, USA); (2) Library Preparation—Ion Ampliseq™ Library Preparation on the Ion Chef System + Ampliseq™ Cancer Hotspot Panel v2 Chef-Ready Kit (ThermoFisher Scientific); (3) Automated template preparation, chip loading, and sequencing—Ion 510 and Ion 520 and Ion 530 Kit—Chef + Ion GeneStudio S5 System; (4) Analysis—Ion Reporter and Oncomine Reporter (ThermoFisher Scientific Inc., Waltham, MA, USA).

The NGS results considered to be somatic were the ones that showed an allele frequency below 50%, and any genomic alterations detected that showed an allele frequency above 50% were considered germline [5]. The main somatic genomic alterations both from the histologically normal tissue and the tumor tissue are represented in Table 1 and Table 2. All of the detected genomic alterations, both somatic and germline, are included in the Appendix A. The histologically normal tissue was remarkable for the genomic alterations in the following genes: *TP53*; *ABL1*; *ERBB4*; *PIK3CA*; *KIT*; *EGFR*; *MET*; *GNAQ*; *RET*; *FGFR2*; *ATM*; and *CDH1*. The tumor was remarkable for genomic alterations in the following genes: *TP53*; *SMAD4*; *VHL*; *PIK3CA*; *FGFR3*; *PDGFRA*; *KIT*; *KDR*; *FBXW7*; *APC*; *ATM*; *PTPN11*; *FLT3*; *RB1*; *AKT1*; *CDH1*; *ERBB2*; *STK11*.

The patient was evaluated at 3, 6, and 12 months after the nephrectomy procedure with no signs of cancer recurrence and no signs of metastasis. Presently, at 4 years of follow-up, the patient is well and cancer-free.

## 3. Discussion

In this case presentation, we demonstrate a unique growth pattern of PRCC (WHO 2022) presenting in a 59-year-old male patient and subsequently provide a mutational signature of the histologically normal kidney tissue and tumor tissue, using NGS to further explore the molecular basis of this unique presentation. While many of the growth patterns are described in the literature for PRCC [1,4,7,8], to our knowledge no other case similar to this one is characterized in the scientific literature.

Regarding the surgical treatment of choice, the 2018 EAU guidelines recommend radical nephroureterectomy (RNU) for confirmed UUTC [5]; nevertheless, in our case, the suspicion of urothelial carcinoma of the renal pelvis based on urologic CT and no mucosal abnormalities detected in the presurgical ureteroscopy favored the choice of radical nephrectomy.

Next, after the pathology report, based on the available data and the guidelines used in 2018 [9], the patient outcome was considered excellent. The follow-up was stratified at 3, 6, and 12 months for renal function tests, local recurrence, recurrence in the contralateral kidney, and metastases. At 12 months post-surgical intervention, 6 month follow-ups were implemented, with presently, at 4 years after diagnosis, the patient remaining stable with no abnormalities in the kidney function tests and no signs of tumor recurrence. In our case, the renal mass was approachable for a preoperative biopsy strategy, unfortunately, no biopsy of the tumor mass was performed before nephrectomy as per the internal surgical protocols available at the time of the surgical intervention. While currently the diagnostic accuracy of a preoperative biopsy in suspected renal masses is estimated at around 98%, with less than 1% risk of bleeding, this management strategy remains underutilized [10]. This high diagnostic yield is estimated to solidify, as more technological breakthroughs are implemented, i.e., confocal fluorescence microscopy, specifically for preoperative biopsies in urological cancers, this, however, would only be implementable in urological centers with high resources and solid surgical infrastructures [11,12].

Briefly, considering the normal anatomy and micro-anatomy of the kidney [13], the initial tumor most probably developed at the boundary between the renal cortex and the renal medulla and expanded slowly, first as a tumor nodule and then, upon invading a collecting duct, it grew in a cord-like manner till it reached the papillary duct. This provided the expanding tumor access to, and subsequent growth in, the minor and major calyx, and finally the renal pelvis. Here, constant erosion and shedding of the tumor mass elicited the macroscopic hematuria with which the patient presented in the emergency room. The factual basis for the lack of frank invasion of the tumor into the neighboring normal kidney tissue is probably owing to the fact that this uniquely presenting tumor was a low-grade PRCC [4,7]. Considering the somatic genomic alterations identified in the histologically normal tissue (Table 1), we can regard all of the variants identified as contributors to the eventual formation of the tumor, more so when compared with the data in the literature where, specifically, *TP53*, *EGFR*, *CDH1*, and *MET* mutations can be viewed as the initial drivers for tumor genesis, progression, and invasion [14].

The genomic alterations identified in the tumor tissue are in line with the TCGA published dataset on PRCC [2], specifically the presence of a mutation in *VHL* identified in the tumor can be considered a ‘’marker’’ of type I PRCC (WHO 2016). Interestingly, the presence of mutations identified in the *SMAD4*, *FGFR3*, *PDGFRA*, *APC*, *ATM*, *KIT*, *RB1*, *CDH1*, and *PIK3CA* are all known genes with roles in carcinoma development and progression [15], but with low evidence for a contribution to PRCC [2]. Furthermore, no mutations were detected in the *KDM6A*, *SMARCB1*, and *NFL2L2*, as previously described as representative of type 1 PRCC (WHO 2016) [2]. These results can be explained by the low grade of the tumor, which translates into a low mutational burden or mutations that show weak tumor-driver ability. In addition, the heterogeneity in the mutations identified in our case, and the fact that it did not fit with the reported NGS data, provides further background to the concept of a landscape or wide spectrum of tumors in the PRCC family [4]. To further clarify the blurring lines in the spectrum of PRCC, some interesting markers have been demonstrated, specifically, the biomarkers delineating differential diagnosis, i.e., the microRNA—miR-21 increased expression in tissue, and the risk of metastasis i.e., the insulin-like growth factor II mRNA-binding protein 3 (IMP3) increased expression in tissue [16,17].

Finally, considering the low aggressiveness of the tumor, which can be conceptualized by the fact that the tumor most probably had a long growth with regards to the timeframe of development, and no frank invasion of the neighboring kidney tissue, places this particular PRCC on the low-aggressive part of the spectrum of PRCC.

The limiting part of our case report is that we did not evaluate the presence of copy gains in chromosome 7, which represent the particular signature found in the WHO 2016 subdivision which identifies the mutational background in type 1 PRCC. Another limiting factor for our case report is the lack of a multi-omics approach, specifically, transcriptome sequencing, epigenome sequencing, and non-coding RNA sequencing. These NGS methods would offer invaluable insights into the progression of low-grade, non-invasive PRCC to high-grade, frank-invasive PRCC, and the surrounding normal tissue molecular portrait in association with these neoplasms [18]. This testing methodology could be employed to offer better tumorigenesis and tumor progression biomarkers.

In conclusion, we describe the unique growth pattern of this PRCC from CT to NGS and provide a succinct discussion on the pathological background underpinning this particular presentation.

## Figures and Tables

**Figure 1 diagnostics-12-01904-f001:**
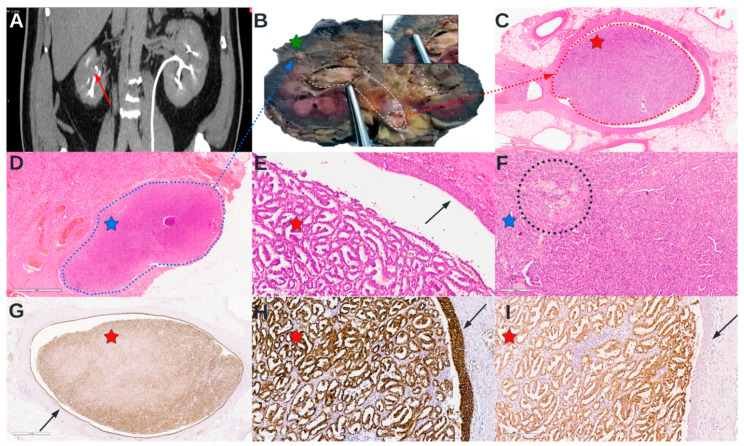
A selected panel of images containing representative snapshots of the tumor from CT to IHC. (**A**) Selected section from the contrast-enhanced CT of the abdomen that showed an hypo-enhancing nodular lesion in the right kidney that measured approximately 38 × 26 × 21 mm (green star on the bivalved nephrectomy surgical specimen while grossing); delayed scan of the lesion that showed the apparent invasion of the renal pelvis (red arrow); (**B**) Pathological grossing of the nephrectomy surgical specimen, bivalved section, that shows a nodular lesion of approximately 3 × 2 × 2 cm located medially, near the corticomedullary boundary; further sections revealed that the nodular lesion was in communication with the cordon-like tumor that shows an intra-pyelocalyceal (non-invasive and compressive) growth pattern (demarcated with dashed white lines), extending all the way to the opening of the proximal ureter; (**C**) H&E section from the renal pelvis showing the cordon-like (demarcated with dashed red lines), red dashed arrow indicates the section origin; (**D**) H&E section from the nodular lesion (demarcated with dashed blue lines), blue dashed arrow indicates the section origin; (**E**) H&E section from the renal pelvis showing the tumor composed of tubulo-papillary structures, the red star indicates the section corresponding to the red star on image (**C**); the black arrow indicates normal urothelium; (**F**) H&E section with higher magnification of the nodular lesion showing compact tubulo-papillary structures lined single layer small cuboidal cells with inconspicuous nucleoli, black dashed circle indicates foamy macrophages; blue star indicates the section corresponding to the blue star on image (**D**); (**G**) IHC for CK7 showing intense and diffusely positive stain both in the tumor and in the urothelium indicated by the black arrow; (**H**) Higher magnification of IHC for CK7 showed in image (**G**), the red star indicates the section corresponding to the red star on image (**G**); the black arrow indicates the urothelium positive for CK7; (**I**) IHC for AMACR showing positive granular stained tumor cells, with negatively stained urothelium marked by the black arrow; the red star indicates the section corresponding to the red star on image (**H**).

**Table 1 diagnostics-12-01904-t001:** Genomic alterations that were detected in the histologically normal tissue. Allele frequency below 50% was considered a somatic mutation [6]. No synonymous variants were included in the table even if the allele frequency was below 50%, but are included in the Appendix A. * Automatic reporting from Ion Reporter and Oncomine Reporter of mutations for the most advanced phase (IV, III, II/III, II, I/II, I) is shown and multiple clinical trials may be available.

Genomic Alteration	Gene	Amino Acid Change	Coding	Locus	Allele Frequency	Variant Effect
TP53 D208G	*TP53*	p.(D208G)	c.623A > G	chr17:7578226	4.10%	missense
ABL1 M244I	*ABL1*	p.(M244I)	c.732G > A	chr9:133738332	11.00%	missense
ERBB4 V167A	*ERBB4*	p.(V167A)	c.500T > C	chr2:212652806	3.08%	missense
PIK3CA M1043 *	*PIK3CA*	p.(M1043*)	c.3127delA	chr3:178952069	0.36%	nonsense
KIT L644P	*KIT*	p.(L644P)	c.1931T > C	chr4:55594228	25.96%	missense
EGFR G575R	*EGFR*	p.(G575R)	c.1723G > A	chr7:55232973	42.57%	missense
MET T1011A	*MET*	p.(T1011A)	c.3031A > G	chr7:116411992	3.83%	missense
GNAQ D236G	*GNAQ*	p.(D236G)	c.707A > G	chr9:80409407	3.20%	missense
RET P785L	*RET*	p.(P785L)	c.2354C > T	chr10:43613890	4.31%	missense
FGFR2 E368G	*FGFR2*	p.(E368G)	c.1103A > G	chr10:123274815	6.94%	missense
ATM S333F	*ATM*	p.(S333F)	c.998C > T	chr11:108117787	25.49%	missense
ATM I1688V	*ATM*	p.(I1688V)	c.5062A > G	chr11:108170497	4.50%	missense
CDH1 T66A	*CDH1*	p.(T66A)	c.196A > G	chr16:68835605	8.70%	missense

**Table 2 diagnostics-12-01904-t002:** Genomic alterations that were detected in the tumor tissue. Allele frequency below 50% was considered a somatic mutation [6]. No synonymous variants were included in the table even if the allele frequency was below 50%, but are included in the Appendix A. * Automatic reporting from Ion Reporter and Oncomine Reporter of mutations for the most advanced phase (IV, III, II/III, II, I/II, I) is shown and multiple clinical trials may be available.

Genomic Alteration	Gene	Amino Acid Change	Coding	Locus	Allele Frequency	Variant Effect
TP53 I254T	*TP53*	p.(I254T)	c.761T > C	chr17:7577520	7.68%	missense
TP53 S241P	*TP53*	p.(S241P)	c.721T > C	chr17:7577560	2.31%	missense
SMAD4 H444W;R445 *	*SMAD4*	p.([H444W;R445 *])	c.1330_1333delCATCinsTGGT	chr18:48603029	22.22%	missense, nonsense
VHL Y156H	*VHL*	p.(Y156H)	c.466T > C	chr3:10191473	3.57%	missense
PIK3CA D84N	*PIK3CA*	p.(D84N)	c.250G > A	chr3:178916863	10.91%	missense
FGFR3 G773S	*FGFR3*	p.(G773S)	c.2317G > A	chr4:1808885	5.32%	missense
PDGFRA R554G	*PDGFRA*	p.(R554G)	c.1660A > G	chr4:55141014	5.76%	missense
KIT	*KIT*	p.(?)	c.1991-2_1991-1delinsC.A	chr4:55595499	3.57%	unknown
KIT	*KIT*	p.(?)	c.1991-2_1991delinsCCC.	chr4:55595499	96.43%	unknown
KDR P479A	*KDR*	p.(P479A)	c.1435C > G	chr4:55972955	28.57%	missense
FBXW7 L594V	*FBXW7*	p.(L594V)	c.1780C > G	chr4:153245411	66.67%	missense
APC L1342S	*APC*	p.(L1342S)	c.4025T > C	chr5:112175316	6.25%	missense
ATM D408E;L409N	*ATM*	p.([D408E;L409N])	c.1224_1227delTCTTinsGAAC	chr11:108119818	13.51%	missense, missense
PTPN11 I494V	*PTPN11*	p.(I494V)	c.1480A > G	chr12:112926860	7.85%	missense
FLT3 L610V	*FLT3*	p.(L610V)	c.1828T > G	chr13:28608228	9.80%	missense
FLT3 N609T	*FLT3*	p.(N609T)	c.1826A > C	chr13:28608230	5.04%	missense
FLT3 E608V	*FLT3*	p.(E608V)	c.1823A > T	chr13:28608233	10.39%	missense
FLT3	*FLT3*	p.(?)	c.1310-2A > G	chr13:28610182	40.00%	unknown
RB1 E554G	*RB1*	p.(E554G)	c.1661A > G	chr13:48955545	10.95%	missense
RB1 Q689R;N690Y	*RB1*	p.([Q689R;N690Y])	c.2066_2068delAGAin sGTT	chr13:49033929	5.17%	missense, missense
AKT1 P42S	*AKT1*	p.(P42S)	c.124C > T	chr14:105246476	3.30%	missense
CDH1 N93S	*CDH1*	p.(N93S)	c.278A > G	chr16:68835687	14.83%	missense
TP53 S20P	*TP53*	p.(S20P)	c.58T > C	chr17:7579855	9.89%	missense
ERBB2 V794M	*ERBB2*	p.(V794M)	c.2380G > A	chr17:37881051	4.85%	missense
STK11 E33K	*STK11*	p.(E33K)	c.97G > A	chr19:1207009	3.15%	missense

## Data Availability

All of the data are available upon request to the corresponding author.

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
