# Peer review of "Unique Growth Pattern Presentation of a Papillary Renal Cell Carcinoma"

_diagnostics, 2022, doi:10.3390/diagnostics12081904_

Round 1

Reviewer 1 Report

The authors presented a detailed description of a pRCC unique growth pattern within a case series setting. The authors provided also a deep genomic analysis considering both normal and tumor tissue. Overall interesting report. I have some minor comments before acceptance.

The authors should consider to discuss the following points to improved the Discussion: surgrical and biopsy strategy including sparing approaches and novel emerging biomarkers that can compliment clinical daily practice (pRCC scenario).

- A recent multicentre experience showed that RAPN was still feasible in the context of locally advanced RCC (Yim et al. doi: 10.1016/j.euf.2020.10.011). Furthermore, the authors should consider to discuss the preoperative biopsy strategy for such a clinical scenario. Renal mass biopsy is still underutilized due to its non-diagnostic rates. As future perspective, confocal microscopy would potentially reduce the amount of non-diagnostic renal mass as found for other urological malignancies (Mir et al. doi:10.23736/S0393-2249.19.03627-0). 

- Considering the novel emerging RCC biomarkers in the context of RCC the authors would consider to discuss the recent ICUD-SIU consultation (doi: 10.48083/XLQZ8269).

Author Response

We the authors would like to show our appreciation by offering our sincere thanks for the very constructive ideas offered. 

We addressed the issues and ideas raised by you and added the manuscript to the suggestions provided. We did not comment on the locally advanced RCC because it is not in the scope of our manuscript. We did however address all the other points raised and further discussed them and of course, referenced them. 

We again thank the reviewer for his/her time and remain ready to further improve our manuscript if it is deemed so. 

The authors. 

Reviewer 2 Report

Harich et al. presented in this report an unique growth pattern presentation of a papillary renal cell carcinoma focusing on genomic alterations potentially influenced alternative growth of PRCC.

Manuscript is well prepared and has potential for publication but needs some revisions.

Abstract - Line 33 and Case presentation - Line 76 - what means renal-ureteral lithiasis - it was on the same site as tumor, presented on the sime time as tumor diagnosis - not clear for readers.

Line 73 and 98 - intrapyelocalyceal - it means localization in both - renal calyx and pelvis - adding renal pelvis or pelvic is unnecessary

Line 79-80 - Radical nephrectomy was performed but in case of UTUC (upper urinary tract carcinoma) is suspect treatment method of choice is radical nephroureterectomy with bladder cuff excision - should be cleared

The discussion lacks the clinical aspect of patient management - whether the form of tumor growth affects the patient management (e.g. scope of surgery, follow-up changes).

Author Response

We the authors would like to show our appreciation by offering our sincere thanks for the very constructive ideas offered and for the subjective corrections. 

We addressed the issues and ideas raised by you and added to the manuscript the suggestions provided, with further clearing for patient management, and further expanded our references to highlight the added changes.

We again thank the reviewer for his/her time and remain ready to further improve our manuscript if it is deemed so. 

The authors. 

Reviewer 3 Report

The paper is well done and planned. Tumor heterogeneity remains the urgent issue in the modern oncology. It is behind the low effectiveness of anti-cancer measures and determines individual sensitivity to therapy. It is believed that genetic profiling of tumors opens up new opportunities in the treatment of patients. Add to the discussion and conclusion your thoughts on the practical application of the results of genome-wide sequencing. 

Author Response

We the authors would like to show our gratefulness by offering our sincere thanks for the appreciation for our manuscript.  
We addressed the idea raised by you, and further highlighted the importance of using NGS technologies for the discovery of new biomarkers for disease detection and progression. 
We again thank the reviewer for his/her time and remain ready to further improve our manuscript if it is deemed so. 
The authors.